# Barriers to Biosimilar Prescribing Incentives in the Context of Clinical Governance in Spain

**DOI:** 10.3390/ph14030283

**Published:** 2021-03-22

**Authors:** Félix Lobo, Isabel Río-Álvarez

**Affiliations:** 1Department of Economics, University Carlos III de Madrid and Funcas, 28903 Getafe, Madrid, Spain; 2Spanish Biosimilar Medicines Association, BioSim, 28027 Madrid, Spain; isabeldelrio@biosim.es

**Keywords:** incentives, clinical governance, biosimilars, Spain, barriers

## Abstract

Incentives contribute to the proper functioning of the broader contracts that regulate the relationships between health systems and professionals. Likewise, incentives are an important element of clinical governance understood as health services’ management at the micro-level, aimed at achieving better health outcomes for patients. In Spain, monetary and non-monetary incentives are sometimes used in the health services, but not as frequently as in other countries. There are already several examples in European countries of initiatives searching the promotion of biosimilars through different sorts of incentives, but not in Spain. Hence, this paper is aimed at identifying the barriers that incentives to prescribe biosimilars might encounter in Spain, with particular interest in incentives in the framework of clinical governance. Both questions are intertwined. Barriers are presented from two perspectives. Firstly, based on the nature of the barrier: (i) the payment system for health professionals, (ii) budget rigidity and excessive bureaucracy, (iii) little autonomy in the management of human resources (iv) lack of clinical integration, (v) absence of a legal framework for clinical governance, and (vi) other governance-related barriers. The second perspective is based on the stakeholders involved: (i) gaps in knowledge among physicians, (ii) misinformation and distrust among patients, (iii) trade unions opposition to productivity-related payments, (iv) lack of a clear position by professional associations, and (v) misalignment of the goals pursued by some healthcare professionals and the goals of the public system. Finally, the authors advance several recommendations to overcome these barriers at the national level.

## 1. Introduction

This paper is aimed at identifying the barriers that incentives to prescribe biosimilars might encounter in Spain. Incentives were chosen as one of the main policy actions to stimulate biosimilar use in Spain because they have an important potential leverage, they are relatively underdeveloped in Spain, and they may be controversial in promoting prescription patterns.

We particularly focus on incentives in the framework of clinical governance as they are intertwined concepts. Clinical governance would be impossible to implement without incentives, and incentives, if not impossible, would be difficult to establish in different frameworks.

Biosimilar medicines significantly help to improve patient access to biological therapies that have revolutionised the prognosis of multiple serious diseases, while contributing to price competition in the market and the sustainability of healthcare systems. If one of the main current problems in health policy is to make access to new medicines compatible with sustainability, biosimilars are part of the solution by freeing up very considerable resources [1] (p. 7). 

Biosimilar medicines have a long history in the Spanish pharmaceutical market. Since the approval of the first biosimilar medicine in 2006, within the European regulatory framework [2,3,4,5,6], there are currently 54 authorised medicines (42 effectively marketed) for 17 active ingredients [7]. Biosimilar uptake varies greatly between Autonomous Communities, hospitals, and clinical services. There is also high variability in uptake between molecules regardless of their time on the market [8]. Even so, a budget impact analysis published by the end of 2020 quantifies the savings generated by biosimilars in the National Health System (NHS) at over €2300 million over the 2009–2019 period: “This shows how the entry of biosimilars into the Spanish pharmaceutical market has led to unquestionable and significant savings, especially in hospital pharmacy” [9] (p. 11). The same study estimates that unless major changes occur in market behaviour, the expected savings for 2020–2022 would exceed €2800 million.

However, the uptake of biosimilars in Spain is below the European neighbouring countries’ average. This is observed in the antiTNF group, epoetin and human growth hormone, three of the six active ingredients for which there are data available, which means that there is room for improvement in their use [8]. Thus, the aforementioned budgetary impact analysis estimates that if biosimilar uptake reached 80% in 2022, the €2800 million would be increased by an additional €430 million. The French government set a similar objective in its National Health Strategy for 2018–2022 [10].

The promotion of biosimilars is part of most Pharmaceutical policy strategies. The Independent Authority for Fiscal Responsibility (Autoridad Independiente de Responsabilidad Fiscal, AIReF), which is responsible for ensuring the sustainability of public finances, considers the promotion of biosimilars the most relevant tool for controlling hospital pharmaceutical expenditure [8]. This institution suggests the establishment of biosimilar prescribing incentives to maximise this savings opportunity. 

The “Action Plan to promote the use of market regulating medicines in the National Health System: biosimilar medicines and generic medicines” of the Ministry of Health states that “In the Autonomous Communities […] actions will be carried out to link financial or other incentives” [11] (p. 38). This two-level (national and regional) approach is because the Spanish NHS is a decentralised system since health competences are transferred to the 17 Autonomous Communities. Coordination, strategy for pharmaceutical policy and medicine pricing and financing decisions, among others, lie essentially with the Ministry of Health, and with Autonomous Communities when it comes to budgeting, purchasing and provision [12].

Further, the Commission for Social and Economic Reconstruction of the *Congreso de los Diputados* (the lower Parliament chamber) dealing with the reform of the NHS to tackle with the Covid-19 pandemic included in its report the need to “significantly increase the proportion of biosimilars” [13] (p. 25). 

In short, the promotion of biosimilars in general, and the establishment of prescribing incentives in particular, seem to be unavoidable tasks according to decision-makers and policy makers in the short-term. 

Therefore, this paper is aimed at identifying the barriers that a model of incentives to prescribe biosimilars might encounter in the context of clinical governance in Spain. Both concepts are intertwined as incentives are the instrument and clinical governance the organisational form. 

This is a pioneering approach, as the research literature on this topic is very scarce.

This work is based on a broader study of the incentives that, in the context of clinical governance, can lead to greater use of biosimilar medicines in healthcare [14]. This study reviews and presents the most outstanding experiences of this sort developed in high income countries and examines the possible barriers to their implementation in Spain. 

## 2. Incentives and Clinical Governance

Incentivising health professionals, especially prescribing physicians, is a crucial issue for the organisation and reform of the NHS and for policies to promote biosimilars. Payment systems, including pay for performance, and competition, including benchmarking and yardstick competition, are typical financial incentives [15,16]. However, the incentives that move people are not only financial, nor, of course, only monetary. In the health sector there are other very powerful motivations such as: dedication to service, altruism, professional satisfaction, and reputation; scientific curiosity, the feeling of belonging to a group, etc. [17]. 

Before moving on, it should be noted that in this study we refer to a very narrow definition of incentive. Specifically, financial incentives that are not necessarily monetary or exclusive to the prescriber. This is critical since many studies about European biosimilar landscape refer to “incentive policies” that are not necessarily financial incentives. For instance, educational campaigns, quotas, or tendering practices are considered incentive tools to increase the uptake of biosimilars and financial incentive would be just another mechanism for that purpose [18,19,20,21]. An important study carried out by the European Commission considers that one of the challenges for the Spanish NHS in the future is “to align the incentives of the different service providers with the system’s quality and efficiency goals. For example, staff incentives could be improved.” [22] (p. 253).

The use of incentives to influence physicians’ prescribing patterns and encourage their alignment with organisational goals is a policy that has been embodied in various experiences over time and across countries. In Spain, towards the end of the 1990s, financial incentives related to prescription were already applied in several Autonomous Communities. By 2018, there were at least seven autonomous communities applying them. The AIReF, in its 2018 review of public expenditure on medication dispensed through prescription, recommends establishing prescription incentives [23]. The same recommendation is made in its recent evaluation of the pharmaceutical spending in the public hospital setting in Spain, but now directly linked to biosimilar prescription: “in view of the success of international experiences, it is proposed to implement a gain-sharing incentive system for hospitals, care services and health professionals” [8] (p. 89). A gain-sharing incentive system (also called gainshare agreement) is based on sharing savings associated with more efficient use of medicines at the same time as any efficiencies made will be invested back into patient care to improve their health outcomes [24].

However, when it comes to promoting biosimilars, there are doubts about the most appropriate type of incentives. Some voices are in favour of financial incentives and argue in their support, for example, their contribution to the progress of biosimilars in Germany. “Prescribers need confidence in outcomes, and they and/or the health system need to benefit financially from using biosimilars” [25]. Other opinions consider that it is better to motivate physicians through schemes that avoid direct financial incentives [26].

Incentives are easier to implement in well-organised broader contexts such as health services following the lines of what is known as “clinical governance”. We acknowledge that there is no consensus definition for clinical governance. Our vision of the concept is as follows: This is an organisational form of health services at the micro-level, aimed at achieving better outcomes in terms of patient health, characterised by the following elements:Involvement of health professionals not only in treatments but in the whole management.Decentralisation of decisions and autonomy of services.Restructuring of services in a multidisciplinary manner aimed at the management of high-quality clinical processes.Measurement and evaluation of performance and remuneration that may include monetary and non-monetary incentives.

Some biosimilar prescribing incentives have been put in place in Europe. Although it is not the scope of this research, we summarise in Table 1 the more relevant initiatives to our view, the British experiences being those closer to our approach of clinical governance.

The gainshare agreements reached with regional or local leadership in the United Kingdom [32,33,34], mostly between 2015 and 2017, under a well-established framework [35] are examples of what we mean by biosimilar prescribing incentives in the context of clinical governance. However, in 2018, NHS England began to link the concept of best-value drug to drug procurement as part of a wider strategy to increase savings [36,37]. While this measure might be effective, it does not fall under our definition. 

In Spain, a good example of combining incentives and clinical governance is the “Área del Corazón” (Heart area) of the University Hospital of La Coruña, coordinated by Dr Alfonso Castro Beiras in the 1990s [38]. The project was based on the willingness to cooperate from the cardiology and cardiac surgery services. This clinical management model was based on four elements: (i) Process standardisation; (ii) Strengthening of information systems; (iii) Use of diagnosis-related groups as patient classification systems and (iv) Self-evaluation. The management of the human and material resources and the control over the budget appear to be decisive for the development of this autonomy-based model. The results were very positive. Activity and care indicators improved, and savings were invested back in human resources, making it possible to staff the new intermediate care unit.

Although this initiative is no longer running, there is no doubt that clinical governance offers good possibilities for the efficient use of effective and good quality biosimilars by prioritizing health outcomes, motivation, quality of care processes and efficient use of resources. Actually, we might be talking about one of the first gainshare agreements in Spain. This precedent seems significant enough to support a pilot gainshare agreement in Spain like those successfully implemented in the United Kingdom [34]. 

It is now time to ask what are the barriers and difficulties that incentives to prescribe biosimilars encounter in Spain.

## 3. Barriers According to Their Nature

As we are particularly interested in barriers to incentives in the framework of a model of clinical governance and both questions are strongly related, we present here different barriers encountered in Spain that we have been found relevant for both concepts.

### 3.1. Payment System

The current payment system is also a barrier primarily for incentives but also for clinical governance. Its regulation, tradition, and the culture it has generated are very much in opposition to the incentives and flexibility required by efficient organisations. The 2006 report on Spain by the European Observatory on Health Systems and Policies noted that after having completed the healthcare transfer process to the Autonomous Communities in 2002 “the most concerning issue was that most of the pay increases affected the fixed components… compared with income related with performance” [39] (p. 110). Several sources suggest that remuneration based on results and effort is motivating and that a variable remuneration based on targets should be increased in relation to fixed payments [40,41,42].

### 3.2. Rigid and Bureaucratic Budgetary and Economic Management Legislation and Procedures

In Spain, bureaucratic control has traditionally prevailed over the evaluation of outcomes, at all governmental levels. “The budgetary system has long suffered from being excessive rigid” [43] (p. 278). These rigid legislation and budgetary and financial management procedures involve major difficulties and delays and are a significant impediment when implementing incentives and clinical governance. The production of health services requires agility and flexibility to manage personnel and material resources to improve health outcomes. Furthermore, as suggested by Zornoza Pérez [43] (p. 295), “flexibility and control must be properly combined with accountability to induce managers to behave in accordance with the principles of efficiency and economy in the management of public expenditure.” This need remains outstanding. In 2016, Esteban and Arias [44] (p. 98) state that “one of the main challenges for the NHS is to progress towards the de-bureaucratisation of the system by leaving the current public law regime in the field of human resources.”

### 3.3. Spanish NHS Labour Relations Model

The employment relationship of physicians and other health professionals with the services that make up the Spanish NHS (so called “statutory personnel”) follows the civil service model. The rigidity and the difficulties it entails to achieve efficient management have often been criticised [45]. One of its main problems is the inflexibility in adapting to care needs and the limitations in differentiating individual and collective merit [46]. It hinders decisively the introduction of incentives and clinical governance. The elimination of this model and the establishment of a modern, flexible, and efficient labour relations system, particularly for physicians, is considered one of the basic structural reforms to be addressed in the NHS. That would mean to eliminate civil-service-like regulations and re-introduce professionalism and evolve towards forms of market labour relations [40,46,47]. 

### 3.4. Clinical and Health Service Disintegration

The disintegration of health care in the NHS is an especially important barrier to implementing clinical governance and promoting biosimilars through incentives. Disintegration implies gaps and borderlines, multiplicity of providers, uncoordinated services, neglect of patient preferences, poor measurement of relevant outcomes, and lack of incentives oriented towards the provision of comprehensive care [48]. Clinical integration is the basic goal of reform plans for the health services to respond to current needs, mainly determined by chronic and degenerative diseases [45,48]. Sometimes integration is accompanied by financing schemes that cover all health services and generate incentives for efficiency. Excessively rigid boundaries between specialties also prevent cross-sectional and team work, organisation of services according to care processes and patient orientation [47]. This is a key difficulty that opposes the development of incentives for individual and collective merit within the framework of clinical governance. 

### 3.5. Absence of General Legislation on Clinical Governance

In Spain, clinical governance is not regulated by law. The draft Royal Decree RD /2015 laying down the basis for the implementation of Clinical Management Units in the Health Services [49] was an interesting initiative. In the interests of a high-quality, safe, and integrated healthcare, the draft suggested providing professionals in the NHS with greater levels of autonomy and responsibility in their clinical decision-making. It included governance-related terms such as planning, incorporation of new technologies and knowledge management, all from a perspective of “decentralisation of the organisation” as well as continuous evaluation of results. Although promising in terms of progress, it was finally rejected by the Council of State because it had to be passed as a law by Parliament. The opposition from trade unions and certain professional spheres, the lack of sufficient support from the Autonomous Communities and the political instability of the past years made it difficult for the draft Royal Decree to become a Law. However, in view of the impact of Covid-19 pandemic, some proposals of the Commission for Social and Economic Reconstruction of the Congress of Deputies suggest that progress could be expected in this regard. “The professionalisation of the governance of health services must be guaranteed and health professionals must be encouraged to perform managerial roles. The executive directors of health services should follow epidemiological, public health and clinical governance approaches” [13] (pp. 3–4). 

### 3.6. Governance-Related Barriers: Lack of Professionalisation of Health Services Managers and Absence of Governing Boards

In Spain, healthcare managers, such as hospital general managers and others, do not always have the appropriate professional profile and appointments are generally based on discretionary decisions. Open and competitive recruitment and selection processes and periodic performance evaluations are not always the rule. Collective and independent boards of directors controlling the micro-management of elementary organisations such as hospitals and health areas in a decentralised and transparent manner [47] are scarce. Only some Autonomous Communities, such as the Comunidad de Madrid [50], have adopted legislation that reflects these principles. In this scenario, major organisational reforms, such as clinical governance and the establishment of incentives, seem unlikely. 

The second Amphos Report [51], prepared with the contribution of 80 managers and clinicians, provides an interesting overview of the barriers that delay the implementation of clinical governance units. Fifteen were identified and classified according to their nature into: political, economic, legal, technological, and human or cultural (Table 2). Some of them match those highlighted above. We also find it interesting to highlight the difficulty in making an organisational change that generates medium-term results when policies are focused on the short term and lack of evidence on objective and reliable results that show the benefits of clinical governance units. These barriers were also classified according to priority. To this end a group of 72 health professionals and managers with previous experience in clinical governance was asked to grade each of the barriers on a scale of 0 to 5 (being 5 the highest) not in absolute values but in comparison with others. The highest barrier was the labour framework, followed by the lack of political will and the regulatory framework. 

The potential savings due to biosimilar competition expected in Spain for the period 2020–2022 (€2800 million) [9] may become the pretext to put in place mechanisms to overcome these barriers, provided, of course, that all parties benefit from these savings. Again, gainshare agreements emerge as a powerful tool for that.

## 4. Barriers According to Stakeholders

### 4.1. Physicians with Limited Information and Distrust of Biosimilars

As biosimilars are biological medicines, they must be prescribed by their brand name [53]. In addition, in Spain the pharmacist cannot dispense a brand other than that prescribed, without authorisation of the prescribing physician according to Order SCO/2874/2007 [54]. Therefore, the physician is the key actor for the market entry of biosimilars. The physician’s trust and preference for prescribing biosimilars is critical.

However, as previously pointed out by Acha and Mestre-Ferrándiz (2017) [55] (p. 263) the biosimilar market faces “the second translational gap” once concerns about the guarantees of the regulatory framework have been dispelled. The authors recognise that “Despite many efforts by regulators to reach out to clinicians, there remains a translational gap for biosimilars which need to be incorporated in healthcare pathways and understood by clinicians and patients. Only by bridging this gap will biosimilars fully play their role in healthcare for Europe”.

Weise et al. (2012) [56] listed the main uncertainties of physicians about biosimilar medicines: (i) doubts about their quality and manufacturing process; (ii) the “similar but not identical” paradigm; (iii) immunogenicity; (iv) possible gaps in post-marketing pharmacovigilance; and (v) extrapolation of efficacy and safety data without clinical trials in certain indications. A recent systematic review conducted on physicians’ perceptions about the uptake of biosimilars suggests that little has changed since then [57]. Physicians still have doubts related to the safety, efficacy, and immunogenicity of biosimilars and consider that cost savings is the main advantages of biosimilars. In addition, most of the physicians had negative perceptions of pharmacist-led substitution of biological medicines.

Another review aimed at identifying the barriers to the use of biosimilar medicines in Europe medicines points out that physicians act as a barrier to biosimilars in several ways [58]. Firstly, their concerns about true similarity between originator and biosimilars. Second, the absence of incentives or benefits for prescribing lower-cost medicines that would compensate the effort they make when explaining to the patient the switch to a biosimilar medicine. Finally, the strong ties between physicians and the originator pharmaceutical companies, which often support clinical research and continuous education (opposite incentive). Given these scenarios, national policies on biosimilars have focused on improving physicians’ trust in biosimilars through a variety of training programs, which are sometimes local and generally relying on prescription guidelines [18].

In Spain, Agustí y Rodriguez (2015) [59] predicted that the success in biosimilar adoption would depend largely on the trust of health professionals and pointed out that the accumulated experience with biosimilars would help overcome reluctant attitudes. 

As biosimilar medicines are mainly used into the hospital setting, hospital physicians have been the focus of many educational programmes (funded by the industry, scientific societies, professional associations, and regional governments). This is also observed in the constant review of position papers on biosimilars by scientific societies in most relevant specialties, such as Oncology, Rheumatology, Haematology, or Digestive Pathology [60,61,62,63,64]. Although one might expect that these statements build trust and shape prescription patterns equally among physicians, a high variability is found when comparing biosimilar uptake between hospitals within a single region. For instance, in 2014, several hospitals in the Community of Madrid rarely used biosimilars, while others showed uptake rates between 60% and 70% [65]. This suggests that despite sharing guidelines from the same regional health service or scientific society, the influence of opinion leaders or heads of departments can accelerate or slow the biosimilar access to hospitals. Nor can we overlook the effect of some sort of incentive set internally at the hospital level, although there is no evidence in this respect.

In the case of primary care setting, the arrival of new biosimilars is a new challenge. From a survey with over 700 respondents, it appears that 58% of the respondents do not know the definition of biosimilars and 73% do not know that the handling of biosimilars is not comparable to that of generics, for which in Spain prescription by active ingredient is applied [66]. Moreover, in the primary care setting, a strategy based on education/information, and constant communication with health professionals, succeeds in improving knowledge about biosimilars and changing prescription patterns [67]. By contrast, any initiative to promote biosimilars not agreed upon with physicians is doomed to failure [26].

In short, physicians that are informed through official and reliable sources tend to consider biosimilars as alternatives that are efficient for the health system and effective and safe for their patients and are able to convey the trust needed for preventing the nocebo effect (nocebo effect refers to negative expectations of the patient regarding a treatment that translate into negative side effects or outcomes) and ensuring treatment compliance. However, improving the knowledge about biosimilars and afterwards communicating the information to patients require big efforts by physicians. Therefore, it cannot be overlooked that recognising physicians’ efforts through incentives or other formulas aimed at sharing benefits will guarantee their commitment in the medium term. 

### 4.2. Misinformed Patients and Mistrust towards Biosimilars

A major barrier to the spread of biosimilar medicines is misinformation and mistrust from the part of patients. The complexity of the world of medicines and their names, especially if they are biological products, makes it difficult for patients to have timely information and knowledge of their characteristics and guarantees [19]. It may be particularly difficult to know and be aware that all medicines that are authorised for marketing, whether they are original products or biosimilars, offer the same safety, efficacy, and quality guarantees. This limited knowledge impacts on their willingness to accept the prescription of biosimilar treatments [68]. In addition, patient organisations often have close links with the originator industry, which sometimes finances their meetings and educational activities [18]. 

The difficulty often arises, not for the naïve patients, but for those whose treatment was initiated with the original product and are encouraged to change it for a biosimilar product (switch). There may be differences in the brand name or appearance [69]. In addition, sometimes the inherent variability in the manufacturing processes of biological products can lead to certain characteristics not being totally identical to those of the originator, but this variability is strongly controlled within acceptable limits to ensure there is no relevant clinical impact [70]. Despite that, biosimilars are not well understood by many healthcare professionals and patients, and such a mistrust is exacerbated by negatively biased information disseminated by some parties [71]. 

A study on policies to promote biosimilars in 24 European countries found that educational initiatives aimed at patients were rare. Patients are informed mainly through their organisations, through brochures and letters to explain the switch from originator to biosimilar. It recognises that biosimilar policies should include all stakeholders, including patients, and recommends strengthening educational initiatives through instruments such as question and answer (Q&A) documents [18].

However, a very recent study by Vandenplas and collaborators (2021) [72] emphasised that over the past few years several surveys among European patients have shown a lack of knowledge and trust in biosimilars. In addition, they performed a web-based screening of European Patients’ Forum and International Alliance of Patients’ Organisations on publicly available information about biosimilars and found a high variability among correctness, the level of detail, and the tone when providing information.

The physician–patient relationship is absolutely crucial to overcome these information or mistrust issues. There is no doubt that as long as the physician is properly informed and trusts biosimilars, the patient will follow his or her guidelines.

It should be borne in mind that having accurate information and access to medicines are rights that are widely recognised in different jurisdictions. According to the Spanish legislation the physician must inform the patient. Indeed, according to Article 10 of the General Health Services Law 14/1986 [73], patients have the right to be informed on the health services they have access to and the requirements for their use. According to Law 41/2002, Article 4, on patient autonomy and rights and obligations regarding clinical information and documentation [74] patients have the right to know all available information on any action touching their health, including, at least, “the purpose and nature of each intervention, its risks and consequences”. In addition, the physician must guarantee the fulfilment of this right to information from the part of patients. Article 10 of the General Health Law (LGS) also states that patients have the right “to obtain medicines and medical devices that are considered necessary to promote, preserve or restore their health”.

In Spain, Calleja et al. (2020) [75] identify patient education and involvement in the decision-making process as key points to increase acceptance of biosimilars and counteract the nocebo effect. This is the view of at least eleven patient associations in Spain as embodied in the “Joint statement by physicians and patients on treatments with originator biologics and biosimilars” [76]. Thus, some requests read as follows “Health administrations often lack biosimilars training programmes for physicians”, “The debate on originator biological and biosimilar medicines should be open to the participation of physicians and patients”, “Policies that would make the cost/efficiency principle a systematic argument would not be acceptable”, or “Some administrative decisions could seriously interfere with the normal functioning of the physician-patient relationship”. The last two could be an obstacle to the establishment of biosimilar prescribing incentives from the patients’ perspective.

### 4.3. Unions Opposition to Productivity-Based Variable Remuneration

One of the barriers to the establishment of biosimilar prescribing incentives is the opposition of trade unions to variable remuneration based on outcomes, targets, or productivity. This exists not only in Spain but also in other countries, and in any sector, not only in health. The study by García-Olaverri y Huerta (2011) [77] shows that trade unions defend salary standardisation and oppose differentiation according to the different abilities or skills of workers. These objections are found also in the governmental and health sectors. 

The disagreement clearly appears in the 2014 document of the State Confederation of Medical Trade Unions (CESM, in its Spanish acronym) on clinical governance: “Under no circumstance may incentives be linked to savings over the agreed budget, but rather to the level of compliance with it, and with the care and quality targets established in accordance with the provisions laid down in the management contract.” “This implies that the health service that decides to promote clinical governance must allocate additional funds to pay for these incentives” [78] (p. 10). This position is clearly contrary to the establishment of, for example, gain-sharing programmes which have been successfully established in other countries [34] where part of the savings from increased use of biosimilar medicines revert to the healthcare system itself.

### 4.4. Professional Corporate Bodies, Clinical Governance and Incentives

Professional corporate bodies, especially those of physicians, react positively to clinical governance insofar as it increases their autonomy, responsibility, and decision-making capacity. Other features such as performance assessment, performance-related incentives and transparency and accountability do not generate the same enthusiasm [79,80]. These corporations may defend based on professionalism and technical criteria organisational and management changes that promote their professional practice and the health of patients. However, they also experience the pressure of electoral cycles. Then, they usually oppose structural reforms advocating the interests of less committed colleagues (as if this behaviour were the rule) to get the most votes in their corporation’s elections. 

### 4.5. Physicians and Other Health Professionals Not Aligned with the Objectives of the System

Although it is a very limited group, health professionals not aligned with the goals of the system, poorly committed to the public system, can be a barrier to clinical governance and incentives for good performance in general, and for biosimilar prescribing incentives in particular. Attitudes such as opposition to transparency or to performance assessment leading to differentiated remuneration must be corrected, as they have a very negative effect on the morale of the vast majority of those who are compliant. When it comes to biosimilars, the strong ties that originator companies have with physicians through supporting clinical research or training may influence prescription choices [55]. Additionally, guidelines with an economic rationale intended to deliver benefits at societal level may be badly received by some physicians, who may consider that their professional decisions are challenged [81]. Thus, it seems reasonable that a greater alignment between the medical community and the regulators would help build trust on biosimilar medicines [82].

## 5. Recommendations for Spain to Overcome the Barriers to Implement Incentives for Biosimilars

According to our review we recommend the following actions to overcome the barriers to implement incentives for biosimilars:Efforts to inform and educate physicians on the pharmacological and clinical characteristics of biosimilars should be continued and intensified, always on a scientific basis.Patients should be informed about biosimilars to ensure their trust on medicines that are approved by the regulatory authorities.We recommend informing all types of unions and professional corporations of the improvements that clinical governance schemes including incentives (especially those based on gain-sharing programmes) can bring about for the NHS, patients, and professional practice.Consensus and support from policy makers is required to implement a growing uptake of biosimilars mainly from the Departments of Health but also from the Department of Finance as its endorsement of financial incentive programs might be necessary.In the long run, structural reforms of the Spanish NHS are required to overcome other barriers to biosimilar prescribing incentives in the context of clinical governance. We refer to rigidity and bureaucracy in management; clinical and health services disintegration; NHS labour relations model; payment systems and governance. Nevertheless, we think that in the short run there is room for new limited experiments, particularly with non-monetary incentives and the gain-sharing design, which will incite less opposition.

## Figures and Tables

**Table 1 pharmaceuticals-14-00283-t001:** European initiatives on biosimilar prescribing incentives.

Country	Level	Incentive Program	Description
France [27]	National-Ministry of Health Hospital and retail pharmacies	Instruction no DSS/1C/DGOS/ PF2/2018/42 du 19 février 2018 relative à l’incitation à la prescription hospitalière de médicaments biologiques similaires […].	Hospitals can earn 20% or even 30% of the difference between the public price of the originator and its biosimilars.
Germany [28,29]	Regional-Saxonia Regional physician association and sick fund	“Biolike” initiative. Agreement between KV Westfalen-Lippe and sick fund Barmer.	Physicians who reach a certain biosimilar uptake are eligible to bill additional services for their patients.
Italy [30]	Regional-Campania Regional Health Service	DGR n.66 del 14.07.2016. Misure di incentivazione dei farmaci a brevetto scaduto e dei biosimilari. Monitoraggio delle prescrizioni attraverso la piattaforma Sani.ARP	Centres can earn 50% of the difference between the public price of the originator and its biosimilars to invest in high-cost innovator medicines; while a 5% will be invested back in the centre which generated the savings.
United Kingdom [31,32,33,34]	Local-Hospital Trusts and Clinical Commissioning Groups	Gainshare agreement between the Trust and the Clinical Commissioning Groups (50:50)	Hospitals can earn 50% of the difference between the public price of the originator and its biosimilars that are reinvested in patient care.

**Table 2 pharmaceuticals-14-00283-t002:** Barriers to clinical governance and score according to priority [51,52].

Nature of the Barrier	Score
**Political**	
1. Institutional support: lack of political will to promote decentralised and autonomous management models.	4.6
2. Centralising trends: management oriented towards control, production of rules and regulations, and concentration of activities.	4.0
**Economic**	
3. Short-term results: Clinical governance units (CGU) generate long-term results.	4.0
4. Insufficient budgets: increased demand for care and scarce resources.	4.0
5. Economies of scale: GCU require a minimum critical mass.	3.2
6. Investment in innovation: lack of resources for innovation and improvements.	3.5
**Legal**	
7. Regulatory framework: regulations that hinder organisational change and lack of regulations for CGUs.	4.5
8. Labour framework: regulations that limit the HR policies needed by CGUs.	5.0
**Technological**	
9. Evidence on outcomes: lack of objective and reliable outcomes demonstrating the benefits of CGUs.	3.6
10. Information systems: lack of coverage of information systems and technologies.	4.0
**Human/cultural**	
11. Managers trust: reluctance to delegate responsibilities and risks.	4.3
12. Culture of innovation: the environment does not encourage change or the search for excellence.	4.1
13. Involvement of relevant groups: reluctance to teamwork from different professionals.	4.2
14. Involvement of clinicians: reluctance to taking risks and co-responsibility.	3.9
15. Leadership skills: poor training of future CGU leaders.	4.3

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
