# Peer review of "Barriers to Biosimilar Prescribing Incentives in the Context of Clinical Governance in Spain"

_pharmaceuticals, 2021, doi:10.3390/ph14030283_

Round 1
Reviewer 1 Report
Please find my review attached. My main concern is the relevance of references pre 2018 to the objective of this article.

Reviewer 2 Report
This manuscript sets out to describe barriers for the installation of incentives to promote biosimilar prescription in the context of clinical governance in Spain. The authors examine (i) barriers related to clinical governance in Spain and (ii) barriers to biosimilar use from a multi-stakeholder view.
Overall, the topic is considered to be relevant and of interest to the readership of the Pharmaceuticals special issue on Biosimilars. The objective of the review, exploring biosimilar incentive design and how this can stimulate biosimilar use, is considered valuable. The current organization of the review and the content that is presented does however not fulfil the full potential of the paper. The manuscript will need to undergo structural changes, in terms of both organization and content presented. Currently, the content presented in the body of the manuscript is insufficiently aligned with the title and aims presented in the abstract. As reader, you would expect specific information on biosimilar incentive design. This is however currently not comprehensively described in the text.
Section 3 presents barriers for clinical governance, rather than barriers for incentive design or biosimilar incentive design. The information is presented in a biosimilar a specific manner. The section would benefit from positioning the barriers for clinical governance more in the biosimilar context.
Section 4 presents various barriers to the use of biosimilars in general, rather than discussing biosimilar in the context of stakeholder motivation and incentive design. The section would benefit from more focused information on biosimilar incentive design.
Due to this structuring, the manuscript currently does not fulfil its preamble of discussing (barriers to) biosimilar incentive design in Spain. The abstract indicates that the study identifies barriers to the establishment of incentives for biosimilars in Spain from two perspectives: (i) based on the nature of the barrier and (ii) based on the stakeholder involved. The body of text however rather describes (i) barriers to the establishment of clinical governance in Spain (in a biosimilar a-specific context) and (ii) barriers to biosimilar use in general (non-incentive specific). Information on biosimilar incentive design should be overall expanded in the second section (now a variety of elements is discussed). The link between the first (clinical governance barriers in Spain) and second (biosimilar incentive possibilities in Spain) section should be more clearly described.
The manuscript currently presents mainly policy information from a theoretical point of view (especially so in the section on clinical governance). The manuscript can be improved by contextualizing the topic more in the published scientific literature regarding biosimilar incentive design. Information and appropriate references to related work on biosimilar incentive design should be added. References about initiatives in other European countries would be of specific interest: what are examples of successful biosimilar incentive design in other countries and how could these be translated to the Spanish context? How can incentive design in Spain be informed by previous studies? What type of incentives exist? A short reference is made to France and Germany. It would be interesting to expand on this: which development and examples are available in other countries and can inform Spanish policy initiatives?
The following references are suggested to include:
- Scotland : Plevris, N.; Jones, G.R.; Jenkinson, P.W.; Lyons, M.; Chuah, C.S.; Merchant, L.M.; Pattenden, R.J.; Watson, E.F.; Ho, G.-T.; Noble, C.L., et al. Implementation of CT-P13 via a Managed Switch Programme in Crohn’s Disease: 12-Month Real-World Outcomes. Digestive Diseases and Sciences 2019, 64, 1660-1667, doi:10.1007/s10620-018-5406-8
- England : Chan, A., Kitchen, J., Scott, A., Pollock, D., Marshall, R. and Herdman, L. 2019. Implementing and delivering a successful biosimilar switch programme—the Berk- shire West experience. Future Healthc J. 2019. Jun; 6(2):143-145. doi: 10.7861/ futurehosp.6-2-143.
- England: Chung, L., Arnold, B., Johnson, R. et al. OC-038 Making The Change: Switching to 1 Infliximab Biosimilars for IBD at North Bristol NHS Trust. Gut 2016;65:A22-A23.
- France, example of incentive design: Athémane Dahmouh (DREES), 2019, « Médicaments biosimilaires : l’hôpital, premier vecteur de leur diffusion », Études et Résultats, n°1123, Drees, septembre.
- Stakeholder insights on stakeholder motivation to use biosimilars and different types of biosimilar incentive design: Barbier, L., Simoens, S., Vulto, A.G. et al. European Stakeholder Learnings Regarding Biosimilars: Part II—Improving Biosimilar Use in Clinical Practice. BioDrugs 34, 797–808 (2020)
- Regression model analysis that demonstrated that incentive policies were correlated to biosimilar uptake: Key drivers for market penetration of biosimilars in Europe. Rémuzat C, Dorey J, Cristeau O, Ionescu D, Radière G, Toumi M.
- Study discussing incentives in 10 European Member States: Cécile Rémuzat, Anna Kapuśniak, Aleksandra Caban, Dan Ionescu, Guerric Radière, Cyril Mendoza & Mondher Toumi (2017) Supply-side and demand-side policies for biosimilars: an overview in 10 European member states, Journal of Market Access & Health Policy, 5:1, 1307315,
The manuscript is presented as a Review (selected article type). No information is included on the type of review (is it a narrative review?) and methodology. Authors state in the manuscript: “This is a pioneering analysis, as the research literature on this topic is very scarce.” (line 67-68)’. It would be helpful to explain how the pioneering analysis was conducted or what it entailed. Authors present a categorization of barriers but currently it is unclear how this categorization was obtained (section 3). Also for section 4, it is unclear how the categorization of barriers for biosimilar use in general was derived.
As the authors mention, the manuscript is based on a broader Spanish report about prescribing incentives for biosimilars in the context of clinical governance. (Reference 13, Lobo, F.; del Río, I. Gestión Clínica, Incentivos y Biosimilares. 1st ed. Díaz de Santos, Madrid, Spain, 2020). What was the methodology followed? In the broader report, the authors also provided a comprehensive classification of incentives and detail past European experiences associated to the incorporation of these incentives. It would be relevant for the reader to include this information on past European experiences with the implementation of incentives within Section 4.
The concept of incentives is currently used as a container term in the manuscript. Different types of incentives could be explored and different stakeholder groups could be targeted. The manuscript would benefit from explaining different incentive options, and referring to other European biosimilar incentive experiences. (see suggested references in the above paragraphs)
In section 2. Incentives and Clinical Governance, the authors signal: “Incentives are easier to implement in well-organized broader contexts such as health services following the lines of what is known as clinical governance”, ‘clinical governance opens good possibilities to the efficient use of effective and good quality biosimilars…’. The authors also indicate that important structural reforms of the Spanish NHS are required to overcome barriers limiting the implementation of biosimilar prescribing incentives. These reforms would require to address, in the long-term, aspects such as the rigidity and bureaucracy in management, the clinical and health systems disintegration, the current NHS relations model and the current organization of the payment and governance systems. It would have been interesting to couple this comprehensive analysis of long-term challenges with an analysis of the feasibility of implementing prescription incentives for biosimilars in the near future. Aspects such as which incentive programs are more likely to be implemented based on the current structural organization of the NHS remain unaddressed. This is of interest based on the intention of policy- and decision-makers to incentivize the use of biosimilars within a short time frame (lines 63-66).
It is considered valuable that the authors end the manuscript with specific recommendations for Spain to overcome the barriers to implement incentives for biosimilars (title of section 5). The recommendations presented here are however not specific to overcoming barriers to incentive implementation. It is suggested to make these more specific and as well practical. Currently, it is unclear what the authors put forward as possible incentive routes to explore in Spain: would a financial compensation for biosimilar prescription quota, or rather a gainsharing initiative of interest?
In section 1 (introduction) and section 2 (incentives and clinical governance), it would be beneficial to structure the text differently. Currently, the concepts of ‘incentives’ and ‘clinical governance’ are alternatingly discussed, which can be confusing for the reader. In addition, the definition on clinical governance should be made more clear. The authors may want to acknowledge that there is no consensus definition for clinical governance (Line 101). It is suggested to define more clearly what are the main elements and the implications of a clinical governance system in the context of this study.
Please explain in the introduction why incentive design is chosen as main policy action to focus on to stimulate biosimilar use in Spain. Several policies have been reported as potential drivers of biosimilar uptake and might explain observed differences in biosimilar uptake across EU countries, such as physician and patient trust in biosimilars, pricing and reimbursement policies, and procurement policies. It would be helpful to explain the importance or opportunity for biosimilar incentive design in the introduction.
Overall recommendation
In summary, this is an interesting and informative manuscript, but structural changes are needed to enrich the information on biosimilar incentive design prior to acceptance.
Line-by-line remarks
- Line 36 – ‘high variability between molecules’ likely needs to be ‘high variability in uptake between molecules’
- Line 44 – here it is mentioned that biosimilar uptake is below average compared to other European countries. It would be helpful to explain why it is important to stimulate biosimilar use. In the previous lines, authors explain that substantial savings have already been made, regardless of the low uptake. It would be useful if authors can add some information about the importance to create a sustainable market climate for biosimilars. It could also be useful to explain that driving biosimilar uptake is not a goal in itself from a societal perspective, but is needed to stimulate competition, which in turn leads to the desired healthcare system benefits.
- Line 45 – “This is observed in three of the six active ingredients”. Please specify which ones.
- Line 58 – it would be helpful in the introduction to provide some key information about healthcare organization in Spain. Here for example it would be useful to explain about the Autonomous Communities
- Line 64 – different type of policy avenues can be explored and stimulated in the context of biosimilars. In this manuscript, the choice is made to focus on incentives. It would be useful if authors explained the choice of focus.
- Line 69 – authors mention that literature on the topic is scarce. Is more literature available about incentive design in healthcare in general which can be useful to take into account here?
- Line 102 – the term micro-management might lead to misunderstanding. It is suggested that authors explain what they understand under micro management in this context or select another term. Micromanagement is generally considered to have a negative connotation.
- Line 108 – ‘A good example was the Área del Corazón’. Is this the only example? How has this initiative evolved in time? It would be useful to explain the example: what did the initiative entail? Is it a good case on incentive design from which other centers in Spain can learn?
- Line 117 – what did the draft Royal Decree of 2015 entail? It is mentioned to be an interesting initiative, but it is not clear from the current text why.
- Table 1, first line – Political instead of policies? (See line 185)
- Table 1 and Figure 1 – the link between the barriers specified in Table 1 and Figure 1 is not always evident.
- Figure 1- The barriers to the establishment of CGUs identified by AMPHOS (reference 33) have been classified according to priority. Although the prioritization scoring procedure may be clearly explained in the AMPHOS report, this is not clear from solely Figure 1, e.g. what does the 3-5.25 scoring system mean? What is the professional profile of the participants that have taken part in the prioritization exercise?
- Figure 1 typo in statement 11 ‘labour framwwork’
- Table 1 and Figure 1 are referenced. It is not clear of these figures are part of the author’s own work or reproduced from someone else and if so, if this is done with permission of the original author.
- Line 214 interesting remark, but suggest rewording to clarify the statement as not everyone will understand it now. Maybe it can be explained here that questions about the regulatory evaluation have largely settled now, but that challenges still exist on how to best implement biosimilars and leverage biosimilar competition at member state level.
- Line 256 – Authors may want to consider to include a short definition on the nocebo effect, as not everyone may be informed about this
- Line 260 Misinformed patients and mistrust towards biosimilars – the role of patient advocacy groups and organizations could be described in more detail (lines 268-269). Also, patients are not the only stakeholder group that may be struggling with biosimilar concepts (and biological medicines in general). In the context of biosimilar misinformation, this may be a useful reference: Cohen, H.P., McCabe, D. The Importance of Countering Biosimilar Disparagement and Misinformation. BioDrugs 34, 407–414 (2020). The paper Barbier, L., Simoens, S., Vulto, A.G. et al. European Stakeholder Learnings Regarding Biosimilars: Part I—Improving Biosimilar Understanding and Adoption. BioDrugs 34, 783–796 (2020) provides insight and recommendations on how to overcome stakeholder misunderstanding about biosimilars
- Table 2. It would be more relevant to include a table on possible biosimilar stakeholder incentives and possibly examples in other countries. Table 2 now presents information on topics to communicate to patients, which is considered to be somewhat off topic here.
- Table 2 appears to be a Table from a previously published paper. It is unclear if the table is reproduced with permission of the original author or not
- Lines 325- 328 – ‘The health service that decides to promote clinical governance must allocate additional funds to pay for these incentives. This position is clearly contrary to the establishment of, for example, gain-sharing programmes which have been successfully established in other countries. Would it not be feasible then to introduce any form of gainsharing in the near future? It would be interesting to talk about possible avenues to overcome this barrier (cfr earlier remark). Plus it would be helpful to introduce and explain the concept of gainsharing, and discuss this incentive in the context of other possible stakeholder incentives. Authors mention that gainsharing programmes have been successfully established in other countries. It would be interesting for the reader to expand on this.
- Line 341, Physicians and other health professionals – The role of other participating stakeholders (e.g., patient advocacy groups, pharmacists, nurses, hospital managers) is not included in the manuscript. The authors may consider that this information is not relevant for the review. If this is the case, it would be interesting to explain why.
- Line 350 ‘according to the analysis just made we XXX’ – suggest to reword this sentence, in line of article type (review/..)
- Line 370 please add that BioSim is the Spanish Biosimilar Medicines Association
- Line 341 – section 4.5: it would be of interest in this section to be more specific about what the misalignment between the goal of the system and of the healthcare professional entails. Do ‘informal’ incentives play an important role in Spain (e.g. informal financial contribution, research support … of pharmaceutical companies that steers physician prescribing?)
- Line 348 – section 5: not all recommendations appear to be focused on barriers to specifically implement incentives for biosimilars. What type of incentive could be explored in Spain? Maybe a pilot project could be initiated? It would be the overall suggestion to make the recommendations more specific and aligned with the aim.
- Are there examples of biosimilar incentives already taking place in Spain?
- Minor spelling remarks: (e.g., line 225 ‘point out’, line 316 ‘this e exists’, Figure 1 ‘11. Labour framwwork, 1. Institutional suppor’)
Reviewer 3 Report
Overall, I find the paper pleasant to read.
I hereby list some suggestions for the authors:
- Paragraph 'Incentives and clinical governance'. I would suggest the authors to introduce a table describing the incentives for biosimilar uptake other EU countries (maybe EU 5 countries) have already adopted. Were the incentives financials or not? I would appreciate if you could citate more literature on the incentives EU countries have adopted to have an overview of the current situation in Europe. For Italy, you can find something in this paper: Effective tools to manage biosimilars prescription: The Italian experience.
- Paragraph 'Barriers according to their nature'. Raw 117. Could you please further explain the aim of the draft Royal decree? It is not clear to me what do you mean for 'In Spain clinical governance is not regulated as such'; Raw 150, it is not clear if and when the civil servant model has been abandoned. What step is Spain at? Please explain better.
- Raw 348, reccomendations. I strongly reccommend to improve the reccomendations part. I would make a synthesis of the main barriers and accordingly, list the reccomendations to overcome each barrier. I would furthermore really appreciate if you could make comparisons with other EU countries. How did they overcome these barriers? Are there European/International guidelines to overcome these barriers?
- Please modify Figure 1 to make it more easy to read. Why numer eleven is the first item? Item 'institutional support' miss the final 't'.
Round 2
Reviewer 1 Report
well done on adding more recent references and explaining better
a few words could be placed differently in the sentence, following suggestions:
166; the British experiences being
232; change is to are
233; delete however
437; biological
438-439; biosimilar medicines in Europe
479; patterns
489; insert the clarification of nocebo effect as made in line 583, delete this in 583
576-577; delete is and who
Author Response
We have followed the suggestions made by the reviewer 1. We really appreciate the advice and suggestions to improve the quality of our manuscript.
Reviewer 2 Report
The authors have adequately addressed review comments and revised the manuscript accordingly. The structure and content of the manuscript has been significantly improved.
Author Response
We thank reviewer 2 for such an exhaustive revision of the document. Your comments and suggestions have undoubtedly helped us improve the quality of the manuscript.
Reviewer 3 Report
The authors made all the adjustments I suggested, the paper can be published in the present form
Author Response
We thank reviewer 3 for the advice and suggestions that have undoubtedly helped us improve the quality of the manuscript.